# Oxygen mediated oxidative couplings of flavones in alkaline water

Xin Yang[1], Sophie Hui Min Lim[1], Jiachen Lin[1], Jie Wu[2,3], Haidi Tang[2,3], Fengyue Zhao[4], Fang Liu[4], Chenghua Sun[5], Xiangcheng Shi[2], Yulong Kuang[2], Joanne Yi Hui Toy[1], Ke Du[1], Yuannian Zhang[1], Xiang Wang[1], Mingtai Sun[1], Zhixuan Song[1], Tian Wang[2], Ji'en Wu[2], K. N. Houk[6] ✉ & Dejian Huang[1,3] ✉

*Catalyzed* oxidative C-C bond coupling reactions play an important role in the chemical synthesis of complex natural products of medicinal importance. However, the poor functional group tolerance renders them unfit for the synthesis of naturally occurring polyphenolic flavones. We find that molecular oxygen in alkaline water acts as a hydrogen atom acceptor and oxidant in *catalyst-free* (without added catalyst) oxidative coupling of luteolin and other flavones. By this facile method, we achieve the synthesis of a small collection of flavone dimers and trimers including naturally occurring dicranolomin, philonotisflavone, dehydrohegoflavone, distichumtriluteolin, and cyclodistichumtriluteolin. Mechanistic studies using both experimental and computational chemistry uncover the underlying reasons for optimal pH, oxygen availability, and counter-cations that define the success of the reaction. We expect our reaction opens up a green and sustainable way to synthesize flavonoid dimers and oligomers using the readily available monomeric flavonoids isolated from biomass and exploiting their use for health care products and treatment of diseases.

Flavonoids constitute a class of plant secondary metabolites that are ubiquitous and diverse in structural variations that have broad bioactivity for human health. The flavonoid core structure features a 15-carbon phenyl-chromone motif and is a privileged structure for drug discovery[1,2]. The structural diversity of flavonoids stems from variable substituent positions of the phenyl, variable numbers and positions of phenol groups on the aromatic rings, number and degree of glycosylation, and formation of flavonoid dimers and oligomers. Flavonoids are known to promote human health not only by reducing the risk factors of non-communicable diseases, including chronic inflammation[3], hypertension[4], diabetes[5], cognitive impairment[6], and diarrhea (Crofelemer)[7], but also combating infectious diseases such as anti-urinary tract infection by A-type

proanthocyanidins in cranberry[8] and potential antivirus (SARS-CoV-2) activity of amentoflavone (3′-8-biapigenin) and its derivatives[9], which are active compounds in *Ginkgo biloba* (the oldest tree on the Earth). While flavone monomers are relatively abundant in fruits and vegetables and can be extracted on an industrial scale from agri-food by-products, flavonoid dimers and oligomers are minor components in plant biomass, and it is not economical to obtain them in a large scale from natural sources. Synthetically, catalyzed C-C bond coupling reactions,[10,11] such as Ullman reaction[12], and Suzuki-Miyaura coupling[13] have been employed (Fig. 1) in order to make a limited number of biflavones with moderate overall yields. These high-temperature reactions suffer from major drawbacks due to the usage of toxic heavy metals, wasteful halogens and boronate by-products,

[1]Department of Food Science and Technology, National University of Singapore, 2 Science Drive 2, Singapore 117542, Republic of Singapore. [2]Department of Chemistry, National University of Singapore, 3 Science Drive 3, Singapore 117543, Republic of Singapore. [3]National University of Singapore (Suzhou) Research Institute, 377 Linquan Street, 215123 Suzhou, Jiangsu, China. [4]College of Sciences, Nanjing Agricultural University, 210095 Nanjing, China. [5]Department of Chemistry and Biotechnology, FSET, Swinburne University of Technology, Hawthorn, VIC 3122, Australia. [6]Department of Chemistry and Biochemistry, University of California, Los Angeles, CA 90095, USA. ✉e-mail: houk@chem.ucla.edu; dejian@nus.edu.sg

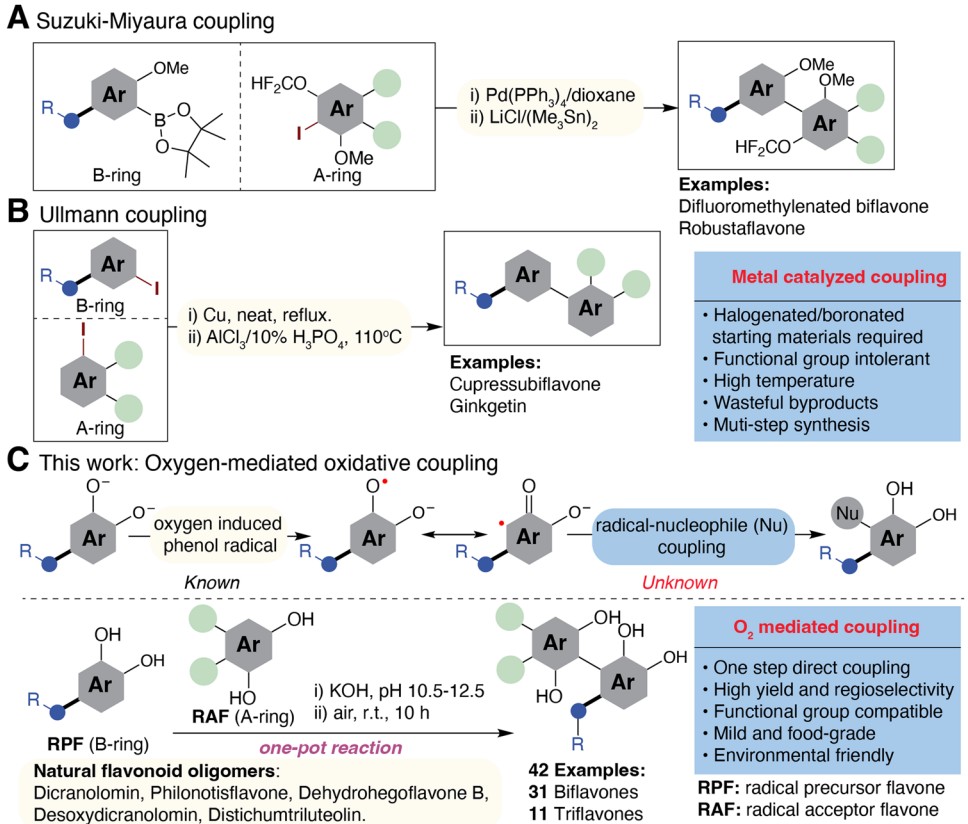

**Fig. 1 | Synthesis of nature-occurring and unnatural biflavones and triflavones.**
**A** Suzuki-Miyaura coupling reaction in the synthesis of biflavonoids. **B** Ullmann coupling reaction in the synthesis of biflavonoids. **C** This work: oxygen-mediated oxidative coupling reaction in the synthesis of flavonoid oligomers; examples include 31 biflavones and 11 triflavones.

and the requirement of protective groups for phenolic functional groups[14–16]. Therefore, these methods fail to meet the stringent requirement of green chemical synthesis demanded by sustainable development[17]. Furthermore, the synthesis of triflavonoids remained as unexplored territory.

Here, we show a *catalyst-free* oxidative coupling reaction of two $sp^2$ C–H bonds of flavones mediated by dissolved molecular oxygen as a hydrogen atom acceptor. Conducted at room temperature and food grade media (alkaline water), our reaction features high yield and good regioselectivity (Fig. 1C). By this simple method, we achieve the synthesis of a large number (>40) of biflavones and triflavones including complex natural products such as dicranolomin, philonotisflavone, dehydrohegoflavone B, distichumtriluteolin, and cyclodistichumtriluteolin found in mosses, one of the oldest land plants.[18] Our discovery is a breakthrough for synthesis and for the exploitation of the great potential of these compounds as pharmaceutical agents and advanced functional materials.

## Results and discussion
### Oxidative coupling of flavones
Hydroxyl group rich flavones (e.g., luteolin) are good reducing agents and have been well-known as potent dietary antioxidants in scavenging biologically relevant reactive oxygen species[19]. Moreover, under alkaline conditions, many weakly acidic flavones including luteolin undergo deprotonation to phenolates, which are sensitive to oxidation by molecular oxygen to their respective ortho-semiquinone anion radicals that have been detected by electron spin resonance spectra[20,21]. Nevertheless, the fates of these radicals are unknown. We envisioned that these electron-deficient semiquinone radicals may react with electron-rich fla-vonoid anions by radical-nucleophile coupling. To verify this, we

conducted HPLC analysis of the alkaline solution of luteolin (pH 11.5) and indeed found several products (Supplementary Fig. 1), which were further characterized as luteolin dimers and trimers by LC-MS. By using the optimal conditions, we scaled the reaction up with 10 g luteolin and successfully synthesized in one-pot **2a** (42%, Lu-(2'-6)-Lu (We use this nomenclature to name flavone dimers and oligomers. For example, Lu-(2'-6)-Lu represents luteolin (Lu) dimer linked by through the C(2') of the first luteolin with the C(6) of the second luteolin), ORTEP plot of its molecular structure is shown in Supplementary Fig. 84), philonotisflavone (**2a'**, Lu-(2'-8)-Lu, 1.2%), dehydrohegoflavone B (**2a''**, Lu-(6'-6)-Lu, 1.0%), and distichumtriluteolin (**3a**, Lu-(2'-6)-Lu-(2'-6)-Lu, 10%) (Fig. 2 and Supplementary Figs. 1–3).

These compounds were originally isolated from moss, one of the oldest land plants, particularly *Rhizogonium distichum* which contains all four triluteolin regioisomers including bartramiatri-luteolin, strictatriluteolin, and rhizogoniumtriluteolin[18]. Their biosynthesis is likely mediated by enzymes (such as polyphenol oxidase) under the neutral physiological pH of moss. High contents of flavonoids in moss (as high as 10% of its dry weight) were suggested to protect the plant from biotic (e.g. fungi) and abiotic stress (temperature, water, reactive oxygen species, and UV-light)[22,23]. Product **3a** was characterized by high-resolution MS, [1]H and [13]C NMR spectra, which reveal the existence of atropisomers due to the hindered rotation of interflavonyl bonds[24], such iso-merism is common in complex natural products including tryp-torubin A[25].

### Cyclodistichumtriluteolin
Triluteolins such as **3a** have one B ring and one A ring on the terminal luteolin units, respectively, that are close to each other for

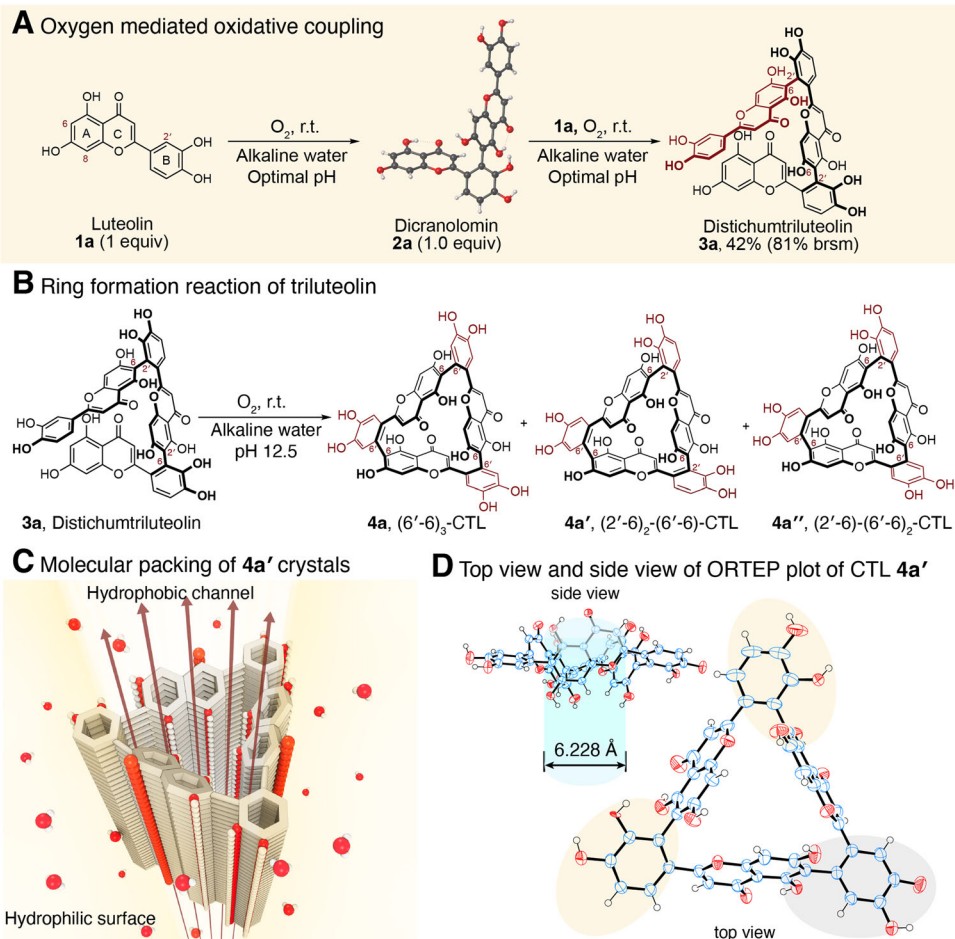

**Fig. 2 | Oxygen-mediated oxidative coupling of flavones. A** Luteolin undergoes oxidative coupling reaction, under weakly alkaline water to give major product dicranolomin (**2a**) and minor products, philonotisflavone (**2a'**, structure not shown), dehydrohegoflavone B (**2a''**) and distichumtriluteolin (**3a**), which could be obtained separately from the coupling of isolated **2a** and luteolin with 42% isolated yield. **B** Intramolecular oxidative coupling **3a** proceeds to give different isomers cyclotriluteolin (**4a**, **4a'**, and **4a''**). **C** Illustration of molecular packing of **4a'** crystals. **D** Top view and side view of ORTEP plot of CTL **4a'**.

intramolecular oxidative coupling (Fig. 2B). By dissolving **3a** in alkaline water (pH 12.5) at room temperature overnight, three major cyclotriluteolins, **4a**, **4a'**, and **4a''**, were formed together with some luteolin monomer and **2a** were observed by HPLC in the reaction mixture (Supplementary Fig. 4). Apparently, interflavonyl bond isomerization has occurred under the reaction conditions and the expected (2'-6)-triluteolin isomer was not detected. The interflavonyl bond cleavage would explain the formation of **1a** and **2a**. These cyclotriluteolins are regioisomers of naturally occurring cyclobartramiatriluteolin ((2'-8)$_3$ interflavonyl bonds) isolated from moss[26]. To confirm the structure of **4a**, we grew single crystals from its methanolic solution and determined the molecular structure shown by the ORTEP plot (Fig. 2C, D), which shows the structure to be **4a'** instead of the expected **4a**. The structure of **4a'** adopts a triangular shape with each corner occupied by B rings of the luteolin and the three edges were fenced by the benzopyranyl moieties (with a length of 7.845 Å) forming a hydrophobic cavity with an opening of ~6.228 Å. The C(4)=O and C(5)-OH form intramolecular hydrogen bond and the benzopyranyl plane tilts with a dihedral angle of about 70 deg with the plane coincident with the paper surface. In the solid state, **4a'** molecules form hydrophobic channels with hydrophilic OH groups (C(7)OH, C(3')-OH and C(4')-OH) pointing outward and C(4)=O and C(5)-OH edge pointing inward. With its unique shape and phenolic groups, cyclotriluteolins may complex guest molecules and metal ions. Thus, it is an intriguing building block for the construction of a functional covalent organic framework (COF).

## Isomerization of cyclotriluteolin

In solution, cyclobartramiatriluteolin exhibited one set of the $^1$H and $^{13}$C NMR spectral peaks for three luteolin units. Due to the C$_3$ axis in **4a**, its $^1$H NMR spectrum agrees with magnetically equivalent luteolin units C(sp$^2$)-H at 25 °C (Fig. 3A). However, due to the presence of three chiral axis along with the interflavonyl bonds, three rotamers are present with equal intensities of $^1$H NMR signals (Fig. 3A). Upon heating the solution to 90 °C in 1 min, interflavonyl bond isomerization occurs rapidly to give new sets of $^1$H signals (Fig. 3B red-colored peaks). We proposed that isomerization could occur through ortho-semiquinone anion radical intermediates facilitated in basic conditions or upon heating (Fig. 3B, C). We calculated the Gibbs free energies of four possible isomers of **CTL** and found that they have relatively similar free energies (Fig. 3D), in agreement with the formation of all these isomers in experiment. This was further supported by the spin-density distribution in CTL radical predicted by DFT (UM062x/6-311+G(d,p)), which shown that both C(2') and C(6') have a comparable spin density (Supplementary Fig. 83).

With the success of homo-cross-coupling of luteolin, we pondered whether a similar homo-cross-coupling reaction could be extended to other flavones. We dissolved apigenin, diosmetin, chrysin, wogonin, 5,6-dihydroxyflavone, and genistein in alkaline water (pH 11.5). However, no desired coupling products were detected under the same conditions. Instead, only starting materials were recovered. No free radical signals were detected by EPR spectroscopy in the reaction solution, suggesting that they are insensitive to oxygen. These flavones

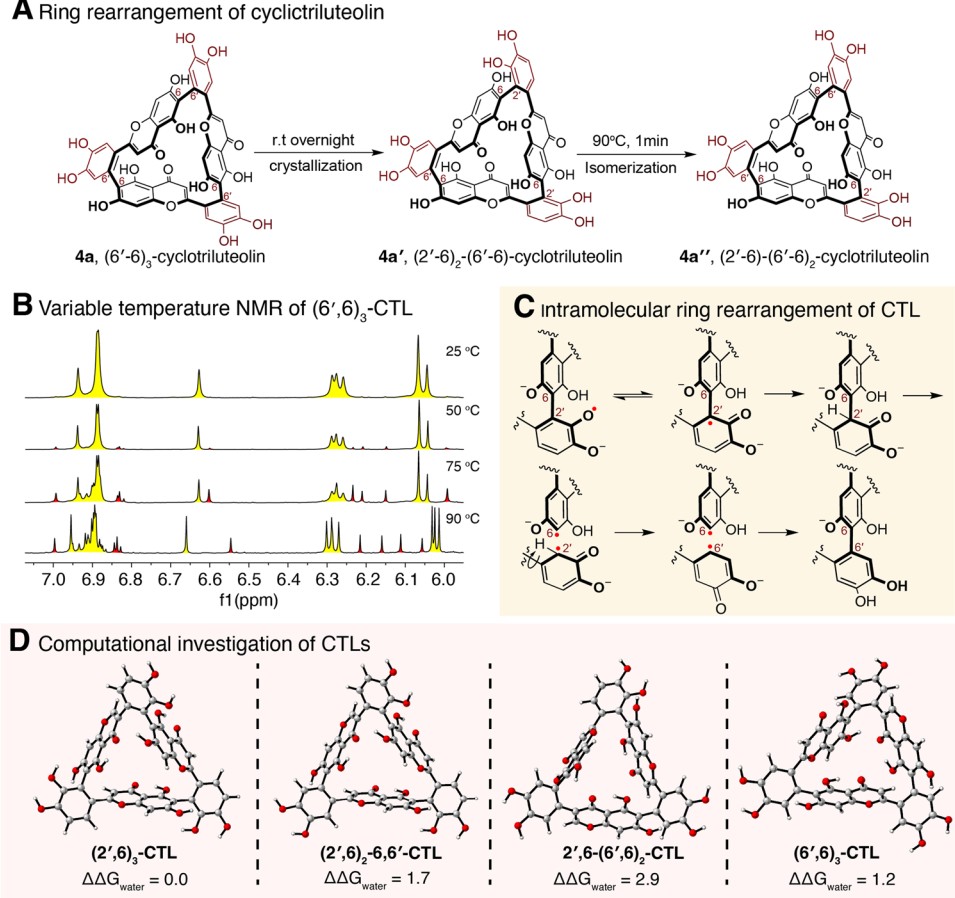

**Fig. 3 | Isomerization mechanism of cyclotriluteolin. A** Ring rearrangement of cyclotriluteolin (CTL). **B** Variable temperature NMR of 4a. **C** Proposed intramolecular ring rearrangement of cyclotriluteolin. **D** Calculated Gibbs free energies of cyclotriluteolin isomers. Calculations were performed at the M06-2X/6-311+G(d,p), SMD(H$_2$O)//M06-2X/6-31G(d)SMD(H$_2$O) level of theory. Energies are in kcal·mol$^{-1}$.

lack catecholic groups preventing them from forming *ortho*-semiquinone anion radicals. In contrast, treating trihydroxyflavones containing catecholic B ring including 3′,4′-dihydroxyflavone, 3′,4′,5-trihydroxyflavone, 3′,4′,6-trihydroxyflavone and 3′,4′,7-trihydroxyflavone in alkaline water resulted in the formation of ortho-semiquinone anion radicals as detected by EPR spectra (Supplementary Figs. 56–59). However, there was little coupled reaction products detected (Supplementary Fig. 5). These observations suggested that these trihydroxyflavones are not sufficiently nucleophilic to accept the semiquinone radicals generated from their oxidation. This agrees with the calculations by density functional theory which found that trihydroxyflavones have much lower nucleophilicity than luteolin (Supplementary Fig. 6). Hence, luteolin anion is unique among these flavones because of its high nucleophilicity and the ability to form *ortho*-semiquinone anion radicals, enabling it to undergo a coupling reaction.

### Hetero-cross-coupling of flavones

When luteolin was mixed with excess apigenin (1:1.5 molar ratio), (Lu-(2′-6)-Ap, **2b**, was isolated in good yield (47%) together with a trace amount of Lu-(2′-8)-Ap, **2b′**, a triflavone (Lu-(2′-6)-Lu-(2′-6)-Ap (**3b**)), and a trace amount of **2a** (Supplementary Fig. 7). The structure of **2b** was confirmed by single crystal X-ray diffraction analysis to be desoxydicranolomin (Fig. 4A), a biflavone isolated from *Plagiomnium undulatum*[27]. Taken together, it became apparent that a general rule for oxygen-mediated oxidative coupling of two flavones is that one flavone is a radical precursor by forming ortho-semiquinone anion radicals, while the other

flavone is a good nucleophile under the weakly basic reaction conditions. This rule is valid for luteolin coupling with diosmetin (Di) (Lu-(2′-6)-Di, **2c**, with 65% yield, Supplementary Fig. 8), chrysin (Ch) (Lu-(2′-6)-Ch, **2d**, 46%, Supplementary Fig. 9), wogonin (Wo) (Lu-(2′-6)-Wo, **2e**, 85%, Supplementary Fig. 10), 5,6-dihydroxyflavone (Df) (Lu-(2′,7)-Df, **2f**, 57%, Supplementary Fig. 11) and Lu-(2′,8)-Df (**2f′**), and genistein (Ge) (Lu-(2′-6)-Ge (**2g**, 28%) and Lu-(2′-8)-Ge (**2g′**, 14%) (Supplementary Fig. 12).

Furthermore, when luteolin was mixed with trihydroxyflavones (TFL) with catecholic group on B ring (**1h-1j**) and 3′,4′-dihydroxyflavone (**1k**, DFL), luteolin became a nucleophile and **1h-1k** were the radical precursor yielding corresponding biflavones FL-(2′-6)-Lu (**2h-2k**, FL = TFL and DFL, Supplementary Figs. 13–16) (Fig. 4A). Not surprisingly, **1h-1k** couple with other nucleophilic flavones such as apigenin, forming FL-(2′-6)-Ap (**2l, 2m, 2n, 2o**) as the sole product (Supplementary Figs. 17–20). These results broaden the scope of our reaction to diverse biflavones containing two different monoflavones. There are many other feasible combinations of flavones and flavone glycosides that can meet this simple requirement.

### Synthesis of hetero triflavonoids

Our reaction can be extended to the synthesis of triflavonoids by reacting **2a** (radical precursor) with nucleophilic flavones. These triflavones share the same type of interflavonyl bonds with the general formula of Lu-(2′-6)-Lu-(2′-6)-FL (FL = apigenin (**3b**), Supplementary Fig. 21) diosmetin (**3c**, Supplementary Fig. 22), chrysin (**3d**, Supplementary Fig. 23), wogonin (**3e**, Supplementary Fig. 24), 5,6-

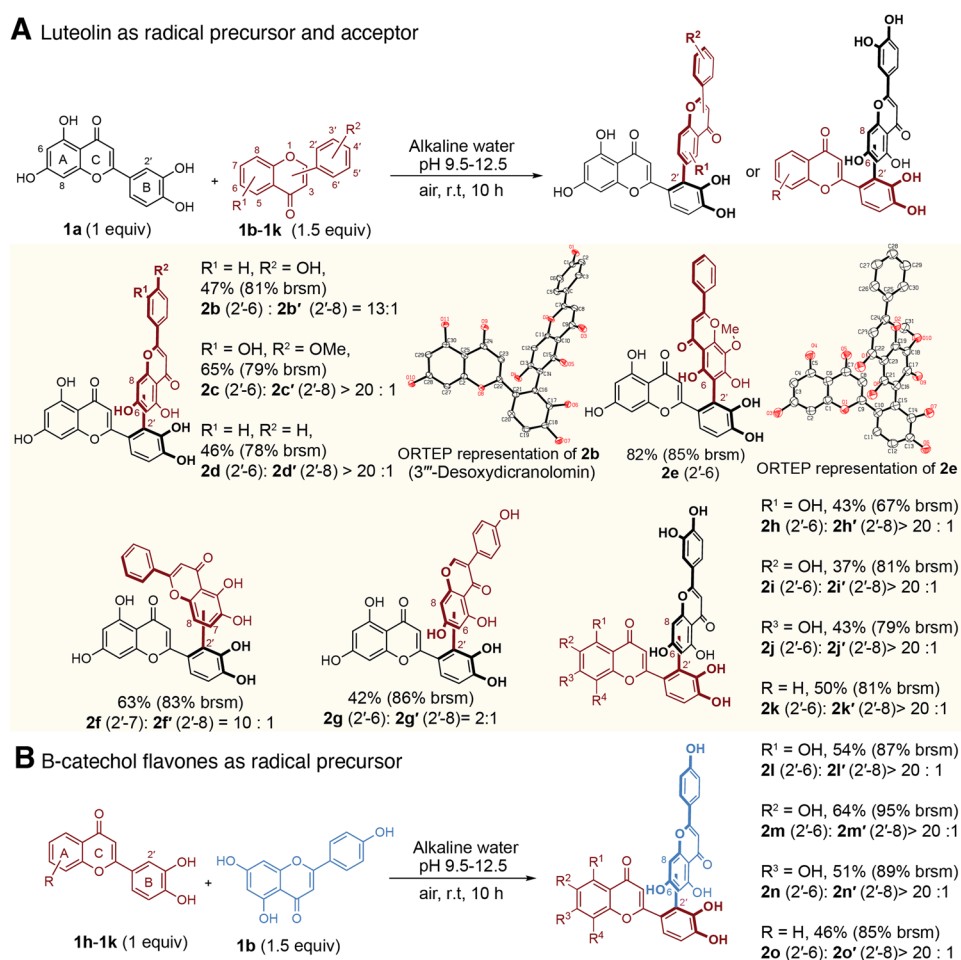

**Fig. 4 | The substrate scope of oxygen-mediated cross-coupling of luteolin and flavones. A** luteolin as a radical precursor and acceptor, and related products, with ORTEP representations for compounds **2b** and **2e**. **B** B-catechol flavones as radical precursors. All yields were isolated and selectivity was determined by HPLC analysis. brsm = yields based on the recovery of starting materials. R = H if not specified.

dihydroxyflavone (**3f**, Supplementary Fig. 25), and genistein (**3g**, Supplementary Fig. 26) (Fig. 5A). The common features for these compounds are the presence of atropisomers due to hindered rotation of the interflavonyl bonds resulting in complex $^1$H and $^{13}$C NMR spectra. Remarkably when **2a** reacted with trihydroxyflavones containing B-catecholic unit, it became a nucleophile and yielded products FL-(2′-6)-Lu-(2′-6)-Lu (**3h**, **3i**, and **3j**) (Fig. 5A. Semi-prep-HPLC chromatograms are shown in Supplementary Fig. 27–29). Biflavones other than **2a** could also be coupling partners allowing the synthesis of triflavones. For example, **2j** has a catecholic unit serving as a radical precursor, and it is coupled with nucleophilic apigenin to form FL-(2′-6)-Lu-(2′-6)-Ap, **3k**, as a sole product (Fig. 5B and Supplementary Fig. 30). **3k** is a unique triflavone containing three different monomeric flavone units. These triflavones all exist as a mixture of atropisomers that could be separated by HPLC (Supplementary Figs. 31–33) but they isomerize over time and show complex $^1$H NMR spectra (Supplementary Fig. 34–45).

## Key factors influencing the reaction outcome

It is well known that under alkaline conditions, flavonoids containing catechol moieties are sensitive to oxidation forming semiquinone radical intermediates. A computational study on neutral flavones also suggested that the presence of catecholic units increases the radical stability through H bonds formation and favors hydrogen atom abstraction.[28] However, the fates of these radicals were unclear and they are not harnessed for synthetic purposes, likely due to the formation of complex end-products. Our discovery is counter-intuitive

and thus warrants an in-depth study on the key factors influencing the reaction outcome so that we can rationally maximize the yield and selectivity for synthetic use. These factors include pH, counter-cations, and oxygen availability in the solution.

**Optimal reaction pH.** Oxidative cross-coupling of two luteolin occurs in a narrow pH range from 9.5 to 12.5, with optimal pH at 11.5 (Supplementary Fig. 46A) and the yield dropped quickly at pH above 13.0. For the cross-coupling reaction between two different flavones, the pH profile is dependent on individual flavones with an optimal pH of 11.0-11.5, except for 5,6-dihydroxyflavone, which has an optimal pH of 10.0 (Supplementary Fig. 46B–J). This observation suggested that the optimal pH of the reaction is determined by the different p$K_a$ values of flavones. Similar pH profiles were found for oxidative coupling reactions between **2a** with flavones (Supplementary Fig. 47). The p$K_a$ values of the luteolin have been reported previously.[29] However, p$K_a$ is highly dependent on solvents and ionic strength. In addition, it is important to determine the positions of deprotonations corresponding to specific p$K_a$ values. Given the fact that luteolin can be oxidized at basic pH, the colorimetric p$K_a$ measurements are subject to interference by the luteolin oxidation products. Therefore, we determined the p$K_a$ values of specific phenolic protons by $^{13}$C NMR spectra of luteolin measured under argon[30] (Supplementary Figs. 48–54). The first deprotonation occurred at C(7)-OH with p$K_{a1}$ of 8.00. This value is about two units larger than literature values (-6.0) (*21*). Our value agrees with the observation that luteolin has poor solubility in water or a slightly basic aqueous solution. The p$K_{a2}$ was found at 8.93 (C(4′)-

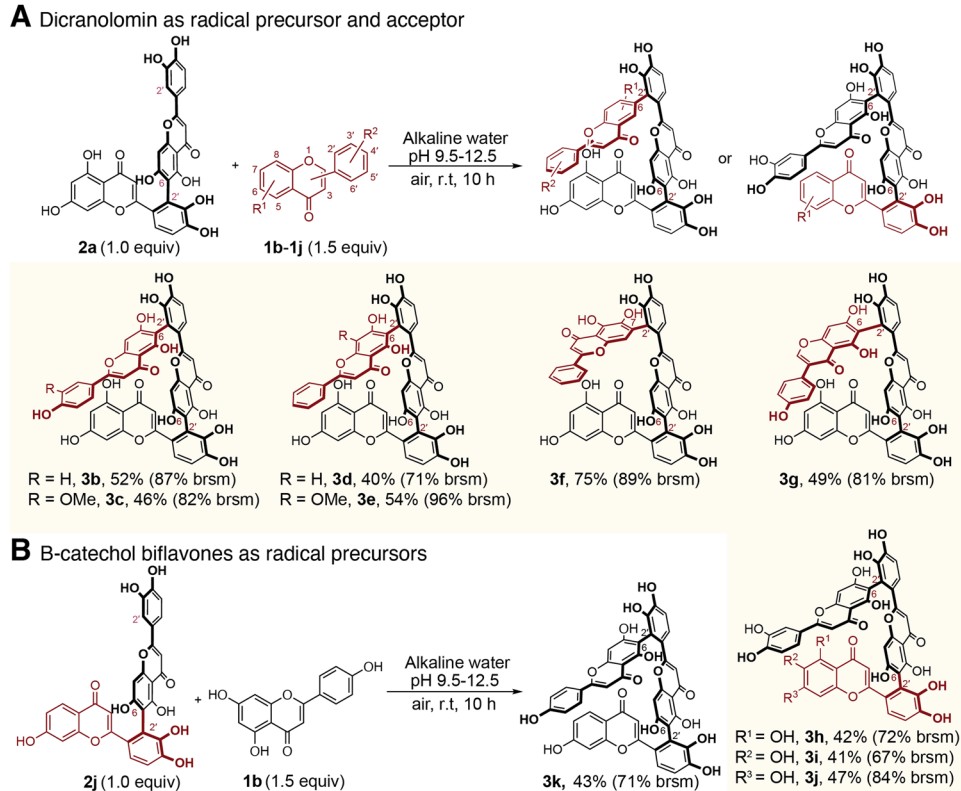

**Fig. 5 | The substrate scope of oxidative coupling for the synthesis of triflavones in alkaline water. A** Dicranolomin as a radical precursor and acceptor. **B** B-catechol biflavone as radical precursors. All yields are isolated yields. brsm = yields based on the recovery of starting materials. R = H if not specified.

OH). The p$K_{a3}$ and p$K_{a4}$ are close to each other at 12.78 (C(3′)-OH) and 13.03 (C(5)-OH), respectively (Fig. 6A). Therefore, under the optimal reaction pH, luteolin (abbreviated as LuH$_4$, instead of Lu to illustrate the degree of deprotonation) dianion (LuH$_2^{2-}$) is the dominant species. To probe the presence of luteolin radical anions, we measured the EPR spectra of air-saturated luteolin solutions in different pH and found that oxidation of LuH$_2^{2-}$ only occurred significantly at pH above 9.5 (Supplementary Fig. 55). This suggested that the oxidation of LuH$_2^{2-}$ can only happen at pH at or greater than p$K_a$ of LH$_2^{\cdot-}$ so that electron transfer-induced deprotonation can occur simultaneously:

$$LuH_2^{2-} + O_2 \rightarrow LuH_2^{\cdot-} + O_2^{\cdot-} \tag{1}$$

$$HO^- + LuH_2^{\cdot-} \rightarrow LuH^{\cdot 2-} + H_2O \tag{2}$$

Thus, the p$K_a$ value of the unobserved intermediate [LuH$_2^{\cdot-}$] dictates the lower limit of the pH range of the reaction. From the EPR signal intensity plots against different pH the p$K_a$ value of LuH$_2^{\cdot-}$ is estimated to be 9.65 (Fig. 7A), which is close to the lower pH limit of the reaction. The LuH$^{\cdot 2-}$ radicals detected by EPR had the same hyperfine coupling patterns (Fig. 6B) that were reported in the literature (21). The spin density of LuH$^{\cdot 2-}$ map shows that C2′ has the highest density (Fig. 6C).

The ortho-semiquinone radicals of other flavones with catecholic B rings were detected, and the C2′s also have the highest spin density (Supplementary Figs. 56–59) in agreement with the fact that C2′ is the major site of the coupling reaction. At higher pH (>12.5), the LuH$^{\cdot 2-}$ radical signal is depleted and a new radical species was detected featuring a doublet of doublets splitting pattern (Supplementary Fig. 60). To pinpoint the nature of the new radical species, the EPR spectra were measured under [17]O labeled oxygen and water,

respectively. We found that [17]O$_2$ did not alter the EPR peak splitting patterns (Fig. 6D). On the other hand, the EPR spectrum of luteolin (pH 12.5) in [17]O-water (30% isotope purity) resulted in a new signal, a sextet due to the hyperfine coupling with [17]O, suggesting that H[17]O$^-$ addition to C$_{2'}$ position (Fig. 6D and Supplementary Fig. 61). We propose that LuH$^{\cdot 2-}$ undergoes dismutation to give an *ortho*-quinone intermediate (Fig. 6E), which can react with hydroxide at high pH resulting in the observed LuOH$^{\cdot 2-}$ radical, which is detrimental to the coupling reaction.

Based on these observations, we propose a coupling reaction mechanism shown in Fig. 7A. In alkaline water, luteolin undergoes deprotonation at C(7)-OH and catecholic protons to give LuH$_2^{2-}$, which undergoes single electron transfer to oxygen, coupled by deprotonation, to give LuH$^{\cdot 2-}$, under oxygen limiting conditions (simply without stirring). This radical anion couples with luteolin dianion, which is the dominant species under the reaction conditions. Computational results suggested that the C(6) of luteolin dianion has lower averaged local ionization energy (ALIE, Supplementary Fig. 82) values than C(8) and thus C(6) is a preferred reaction site, resulting in 2′-6 biflavone as the major product. Furthermore, our computational results also show that 2′-6 isomer dicranolomin (**2a**) is more stable than 2′-8 isomer philonotisflavone (**2a′**) by 1.3 kcal/mol (Fig. 7C). Dicranolomin (**2a**) has two catecholic moieties. Remarkably, its reaction products with flavone monomers are highly regiospecific on the unreacted catecholic moiety (IIB), suggesting that semiquinone radicals from IIB are involved (Fig. 7A). The EPR spectrum of the *ortho*-semiquinone anion radical of **2a** (Supplementary Fig. 62) shows complicated signals due to multiple radical species, including the semiquinone formed on IIB with a$_{HI}$ of 0.285 mT (Supplementary Table 3). The other radical may reside in the IB ring with a$_{HI}$ of 0.43 mT (Supplementary Table 3). The semiquinone radical at IB ring did not participate in coupling reaction, as linear trimer Lu-Lu-FL, instead of the branched trimer, is observed and isolated.

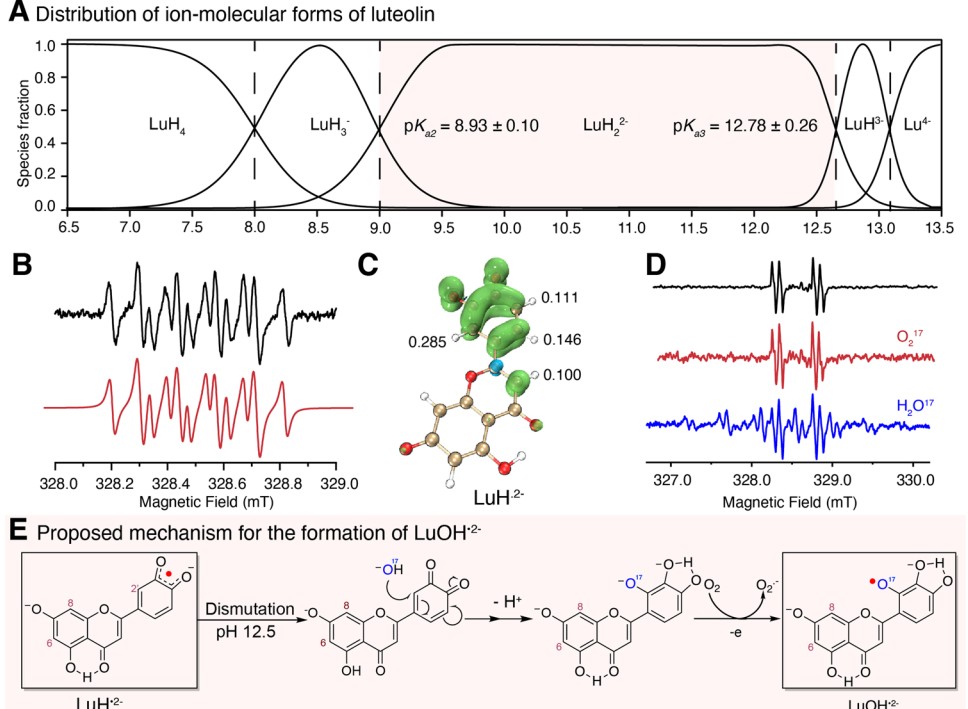

**Fig. 6 | pKₐ profile of luteolin and the radicals generated. A** The distribution curve of luteolin species in aqueous solution LuH$_4$: luteolin, LuH$_3^-$: monoanion, LuH$_2^{2-}$: dianion, LuH$^{3-}$: trianion, and Lu$^{4-}$: tetraanion. **B** Experimental (black) and simulated (red) EPR spectrum of LuH$^{\cdot 2-}$. **C** Spin-density distribution in LuH$^{\cdot 2-}$ predicted with DFT (UM062X/6-311+G(d,p)). **D** Experimental EPR spectrum of LuOH$^{\cdot 2-}$ in air-saturated H$_2$O (Black), in H$_2$O with 30% $^{17}$O$_2$-enriched molecular oxygen (red) and in 30% $^{17}$O enriched water (blue). **E** Proposed mechanism for the formation of LuOH$^{\cdot 2-}$.

**Impact of counter-cations in the coupling reaction.** For two dianions (LuH$^{\cdot 2-}$ and LuH$_2^{2-}$) to reaction, charge repulsions have to be overcome possibly by ion pairing with counter cation. Therefore, we examined the effects of different counter-cations on the reaction; we found that tetramethylammonium (Me$_4$N$^+$, added as Me$_4$NOH) gave the lowest yield (<20%). Lithium performed better but not as good as cesium, sodium and potassium (~80%) (Fig. 7C). These results suggested that the counter-cations not only offset the anionic charges but might also facilitate the reaction through bridging both coupling partners closer to each other with weak coordination interactions. In this regard, a small lithium ion is not as effective as larger alkali metal ions. In aqueous solution, alkali metal ions are present as hydrates, and the coordination bonds with phenolates of the luteolin dianions shall be fairly weak and dynamic.

**Impact of oxygen availability.** Although ortho-semiquinone anion radical of luteolin has been observed previously, the end-products were found to be complex and were not characterized, likely due to overoxidation by excessive oxygen in the solution. The positive outcome of our case is likely due to limiting oxygen and by conducting the reaction unstirred, which is a counter-intuitive result. We compared the reaction dynamics of alkaline luteolin solution in two test tubes; one tube was magnetically stirred vigorously (so that oxygen is in excess supply) while the other tube was not stirred (oxygen availability is dependent on the diffusion of the gas phase oxygen into solution). The coupled products in the stirred tube could not be detected after 4 h, while in the unstirred tube, both **2a** and **3a** showed two major products after 10 h (Supplementary Figs. 63, 64). The air-saturated water has a dissolved oxygen concentration of about 256 μM and its concentration is lower in alkaline water[31]. At the beginning of the reaction, the dissolved oxygen in both tubes was quickly depleted by reacting with LH$_2^{2-}$. However, stirring replenishes the dissolved oxygen, which undergoes radical coupling reaction with LH$^{\cdot 2-}$ leading to

overoxidation. In the unstirred tube, such a reaction is prevented due to depleted oxygen and the slow diffusion of the gaseous oxygen to the undisturbed solution which will prevent product formation over time (Supplementary Figs. 65, 66). Therefore, limiting oxygen availability is a key factor for oxygen-mediated oxidative coupling reaction of flavones.

Other oxidants may also trigger oxidative coupling of luteolin. For example, DPPH (2,2-diphenyl-1-picrylhydrazyl) was able to trigger reaction between luteolin and cysteine ethyl ester to form a low yield 1,4-thiazine derivative of luteolin through 2'-position (B-ring) via sulfur and at the 3'-position via nitrogen[32]. We found that when luteolin was treated radical precursor 2,2'-azobis(2-amidinopropane) dihydrochloride (AAPH), luteolin reacted with radicals generated from AAPH and resulted in isolation of a lactone derivative of luteolin via its B ring (C-2' and C-3')[33]. In both cases, the reactions were likely via a radical-radical mechanism. In contrast, the coupling reactions we reported herein involved A-ring of flavones as a nucleophiles.

**Contrasting bioactivity of the flavonoid dimers and trimers**
It has been suggested that moss utilizes a large amount of bioresource in the synthesis of luteolin dimers and trimers, because of the need to defend against the microbial stress endured by the moss while growing on rotting wood in wet forests[34]. To test this hypothesis, we measured the antifungal activity of luteolin, dicranolomin (**2a**), and distichumtriluteolin (**3a**) using *Aspergillus niger* as a model fungus (Supplementary Fig. 67). Dicranolomin (**2a**) and distichumtriluteolin (**3a**) inhibit the growth of *A. niger* with IC$_{50}$ of 0.86 μM and 0.96 μM, respectively, which is comparable to that of amphotericin B (IC$_{50}$ of 0.50) in a dose-dependent manner. Notably, dimer **2a** shows slightly higher activity than trimer (**3a**). Plant flavonoids protect the plant from being eaten by insects by inhibiting digestive enzymes such as α-amylase and α-glucosidase. We measured the activity of selected

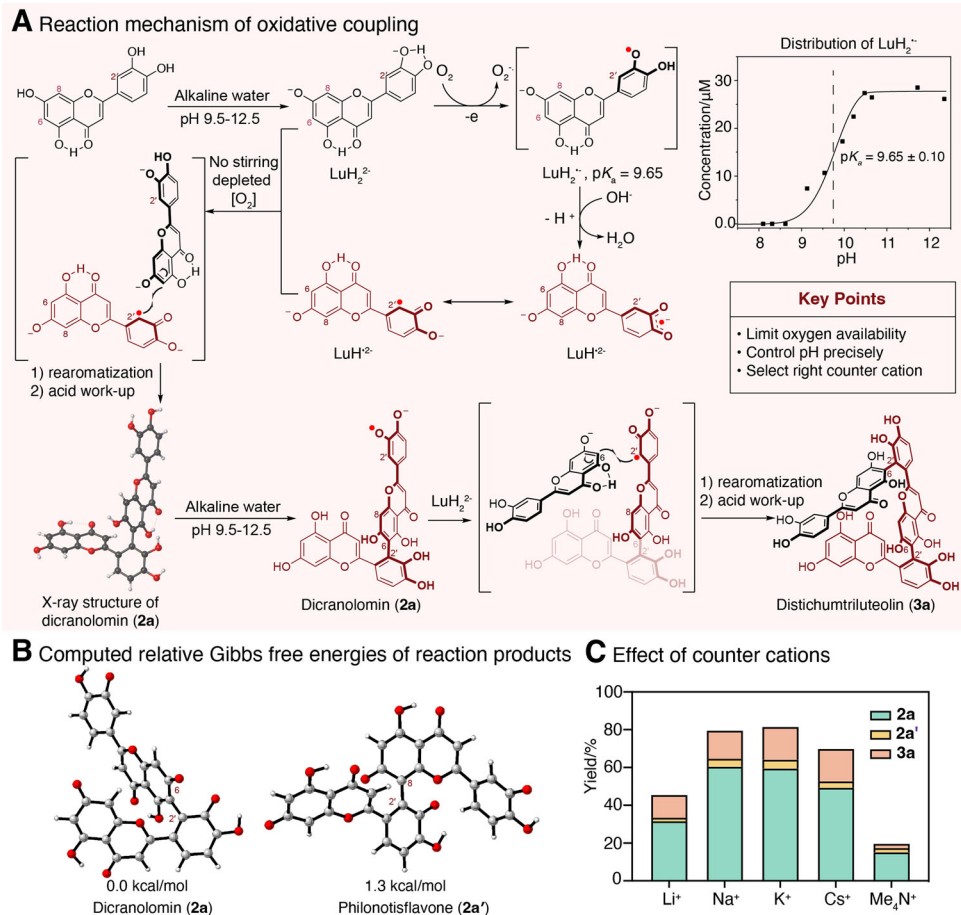

**Fig. 7 | Reaction mechanism and computational study of regioselectivity.**
**A** Proposed mechanisms for oxidative coupling reactions, counter-cations are omitted for clarity. **B** Computed relative Gibbs energy difference between **2a** and **2a′**. **C** The impact of counter-cations on the yields coupling reaction of luteolin.

flavone dimers and trimer (**2a-2f**, **3a-3f**) in inhibiting α-amylase and α-glucosidase (Supplementary Figs. 68–78) and found that dimers **2a** and **2b** show comparable (**2a**) or even higher (**2b**) activity than acarbose, an antidiabetic drug. Molecular docking by using the crystal structure of a pancreatic α-amylase with acarbose complex[35] found that flavone oligomers (**2a**, **2b**, **3a**) are located at the active center while that of luteolin is not, which might explain why luteolin has no activity. The active site region of α-amylase is a V-shaped depression located at the carboxyl end of Glu233, Asp300, and Aspl97. The molecular shape of **2a** and **2b** (Supplementary Figs. 79, 81) happens to be V-shaped (similar to that of acarbose in Supplementary Fig. 80) and fits nicely in depression with one B-ring near the catalytic active groups and form hydrogen bonding through the catecholic group with Glu233, Asp300, and Aspl97.

In summary, we have demonstrated that by judiciously controlling the reaction conditions, the oxygen-mediated oxidative coupling reaction can be achieved for synthetic purpose and that the reaction scope may be extended for the facile construction of other complex molecules by coupling other naturally occurring phenolic compounds such as stilbenoids and auronoids.

## Methods
### Materials
Chemicals and Apparatus. Sodium hydroxide, potassium hydroxide, lithium hydroxide and cesium hydroxide were obtained from Merck & Co., Inc. Flavones including luteolin, apigenin, diosmetin, chrysin, wogonin, genistein, 5,6-dihydroxyflavone, 5,3',4'-trihydroxyflavone, 6,3',4'-trihydroxyflavone, 7,3',4'-trihydroxyflavone and 3',4'-dihydroxyflavone were obtained from Indofine Chemical Co., Inc., Hillsborough, NJ, USA. All aqueous solutions were prepared with 18.2 MΩ·cm ultrapure water obtained by a Millipore water purification system. [1]H NMR spectra were measured using a Bruker AVANCE I 500 NMR spectrometer, and mass spectrometry was performed on a Thermo Scientific LCQ Fleet ion trap mass spectrometer in ES positive mode. The EPR spectra were recorded on X-band EPR spectrometer (JES-TE100, JEOL, Tokyo, Japan), which was equipped with WIN-RAD EPR Data Analyzer System (Radical Research, Inc., Hino, Tokyo).

### General procedures
Reactions were carried out with commercially available reagent in centrifuge tube (50 mL) under room temperature without magnetic stirring unless specified. The isolated yield was calculated from the mass obtained by semi-prep HPLC with C18 column. HPLC yields for optimization of experimental conditions were determined by Waters HPLC analysis system using the corresponding commercial C18 column as stated in the experimental procedures at 25 °C with the PDA detector at 300 nm. [1]H NMR spectra were recorded on Bruker 500 MHz or 400 MHz NMR spectrometers. TopSpin 4.0.5 software was used for NMR data collection and MestReNova v14.2.1 was applied for NMR data simulation and analysis. Chemical shifts were reported in ppm from the solvent resonance as the internal standard (DMSO-$d_6$, δ = 2.50). Spectra were reported as follows: chemical shift (δ ppm), multiplicity (s = singlet, d = doublet, t = triplet, q = quartet, hept = heptet, m = multiplet), coupling constants (Hz), integration and assignment. [13]C NMR spectra were recorded on commercial instruments (126 MHz). Chemical shifts were reported in ppm from the solvent resonance as

the internal standard (DMSO-d$_6$, δ = 39.52). HRMS was recorded on a Bruker Daltonics microTOF Mass Spectrometer (ESI source).

## Separation of reaction products

Isocratic elution method was applied for products analysis of luteolin homo-coupling. DI water with 0.1% formic acid was choose as mobile phase A while acetonitrile with 0.1% formic acid was applied for mobile phase B. The column was equilibrated with 71.5% mobile phase A for 10 min before isocratic elution of the same percentage mobile phase A from 0 to 35 min at a flow rate of 1.0 mL/min, 10 μL from reaction solution was injected into the HPLC system for analysis.

The LC–MS system was equipped with a C18 column (Phenomenex, Luna 5u C18, 250 × 4.6 mm) guard column (4 × 3.0 mm) was applied for product characterizations. The samples (10 μL) were filtered through 0.2 μm membrane (Merck Millipore, USA) before being injected into the HPLC system. The Waters 2998 Photodiode Array (PDA) Detector was connected to the HPLC system with detection wavelengths from 190 and 800 nm. Bruker AmaZon-X is applied for LCMS and LCMSMS for characterization and fragments analysis of unknowns. The LC conditions for LC−MS analysis were similar to those mentioned above. All mass spectra were acquired in both positive and negative ion modes using electrospray ionization. The parent ion was selected with a width of ±2.5 Da and fragmented with 50% setting.

## Synthesis of luteolin-luteolin cross-coupling dimers

**Small scale reaction.** Luteolin (143 mg, 0.5 mmol) was added in potassium hydroxide solution (0.1 M, 30 mL) in 50 mL centrifuge tube. The resulting solution had pH value of 11.50. The tube was closed tightly and place at room temperature overnight without stirring. The solution was then acidified with concentrated hydrochloric acid (1.0 mL, 10 M) to give a solution with pH value of 1–2. Then the solution was extracted with ethyl acetate (3 × 50.0 mL). The organic layer was separated and combined. Removal of the volatiles in vacuo resulted solid, which was separated over semi-prep HPLC with automatic fraction collect system to give the pure 2a, 2a′, and 3a. Their yields and spectral data are described in Supporting information.

**Ten Gram-scale synthesis of luteolin coupling products.** Luteolin (10 g) were weighed then dissolved in potassium hydroxide solution (0.05 M, 2.0 L) in a four 1-liter plastic bottle with 500 mL per bottle. The solution was stand still at room temperature overnight and neutralized with hydrochloric acid (10. M) to give precipitates. The mixture was extracted with ethyl acetate three times (500 mL each) three times. Then the organic layers were combined and concentrated in vacuo to give crude solid product, which was dissolved in methanol and purified over semi-prep HPLC with automatic fraction collect system to give the pure products dicranolomin (2a, 4.2 g, 42%), philonotisflavone (2a′, 0.12 g, 1.2%), dehydrohegoflavone B (2a″, 0.10 g, (1.0%), and distichumtriluteolin (3a, 1.0 g, 10%).

## Synthesis of luteolin-flavone biflavones

Luteolin (72 mg, 0.25 mmol) and flavones (0.5 mmol) were added to an alkaline solution (30 mL 0.1 M KOH) in 50 mL centrifuge tube, pH was adjusted to optimal pH before the tube is closed tight and kept at room temperature for 10 h without stirring. Hydrochloric acid (1 mL 10 M) was added into reaction mixture to quench the reaction. The mixture was extracted with ethyl acetate (3 × 50 mL). The organic phase was combined and evaporated in vacuo. The crude solid obtained was purified over semi-prep HPLC with automatic fraction collect system to give the desired products. The yields and spectral data of each product (**2b-2k**) were stated in section 1.2.8.

## Synthesis of B-catecholic flavone-apigenin biflavones

B-catechols (0.25 mmol) and apigenin (140 mg, 0.5 mmol) were added to alkaline solution (30 mL 0.1 M KOH solution) in a 50 mL centrifuge

tube. The solution pH was adjusted to optimal pH before the tube was closed tightly and kept at room temperature for 10 h without stirring. After that, hydrochloric acid (1 mL 10 M) was added into the reaction mixture, which was then extracted with ethyl acetate (100 mL). The organic extract was worked up and purified over semi-prep HPLC with automatic fraction collection system.

## General procedure for syntheses of triflavonoids

Dicranolomin (143 mg, 0.25 mmol) and flavones (0.375 mmol) were added to an alkaline solution (30 mL, 0.1 M KOH) in a 50 mL centrifuge tube and the pH of the mixture was adjusted to 11.5 before the tube was sealed at room temperature for 10 h without stirring. Acid (10.0 M HCl) was then added into the reaction mixture until the pH reached 1–2. Subsequently, ethyl acetate (3 × 50.0 mL) was added to extract the products, combined organic layer was thoroughly concentrated to give crude solid which was purified over semi-prep HPLC with automatic fraction collect system to give the pure cross-coupling product.

## General procedure for syntheses of cyclotriluteolins (CTLs)

Distichumtriluteolin (50 mg, 0.06 mmol) were added to alkaline water (100 mL, 0.1 M KOH) in a 50 mL centrifuge tube. The pH of the solution was adjusted to 12.5 before the tube is closed tightly and kept at room temperature for 10 h without stirring. After that, acid (10.0 M HCl) was added into reaction mixture until the pH reached between 1 and 2. Subsequently, ethyl acetate (3 × 50.0 mL) was added to extract the products, combined organic layer was thoroughly concentrated to give a crude solid, which was purified over semi-prep HPLC with automatic fraction collection system to give the pure cyclic products.

## Determination of the $pK_a$ values of luteolin by $^{13}$C NMR spectroscopy

Luteolin (200 mg) was added in the three-neck round bottle flask fitted with a pH meter (Supplementary Fig. 50) under argon. Degassed water (20 mL) was added with stirring. To the solution, degassed aqueous potassium hydroxide solution was injected through a syringe to the adjust pH. Aliquots (500 μL) of samples at a given pH value were transferred to 5 mm NMR tubes (Armar Chemicals, Switzerland). To the same NMR tube, deuterium oxide (D$_2$O, 5.0 μL) was added introduced for deuterium lock and a DMSO-d$_6$ solution encapsulated in a capillary tube (Supplementary Fig. 50) was placed in the tube as the external standard for (at 2.50 ppm for DMSO-d$_5$[36] The spectra were recorded and analyzed with MestReNova (Mestrelab research, version 5.2.5−4119). The resulting sigmoidal curve was subjected to nonlinear curve fitting using Prism® (GraphPad, version 5.0b and 5.0f). The dissociation constant $pK_a$ was calculated Henderson-Hasselback equation (Eqs. (1–4)).

$$pK_{a1} = pH + \log_{10}\frac{LH_4}{LH_3^-} \quad (1)$$

$$pK_{a2} = pH + \log_{10}\frac{LH_3^-}{LH_2^{2-}} \quad (2)$$

$$pK_{a3} = pH + \log_{10}\frac{LH_2^{2-}}{LH^{3-}} \quad (3)$$

$$pK_{a4} = pH + \log_{10}\frac{LH^{3-}}{L^{4-}} \quad (4)$$

where LH$_4$, LH$_3^-$, LH$_2^{2-}$, LH$^{3-}$ and L$^{4-}$ are the species of luteolin in fully protonated form, first deprotonated, second deprotonated, third deprotonated 100% deprotonated form.

The chemical shift of the carbons in luteolin is dependent on the relative concentrations of the conjugate pair. For a solution at pH

within $pK_{a1}$ range, and the species in solution is ~100% protonated ($LH_4$), and the chemical shifts are that of the protonated species, $\delta_{low-1}$. For a solution at a higher pH the species in solution is first deprotonated ($LH_3^-$), and the chemical shifts are that of the deprotonated species, $\delta_{high-1}$.

$$\frac{LH_4}{LH_3^-} = \frac{\delta - \delta_{low-1}}{\delta_{high-1} - \delta} \quad (5)$$

$$\frac{LH_3^-}{LH_2^{2-}} = \frac{\delta - \delta_{low-2}}{\delta_{high-2} - \delta} \quad (6)$$

$$\frac{LH_2^{2-}}{LH^{3-}} = \frac{\delta - \delta_{low-3}}{\delta_{high-3} - \delta} \quad (7)$$

$$\frac{LH^{3-}}{L^{4-}} = \frac{\delta - \delta_{low-4}}{\delta_{high-4} - \delta} \quad (8)$$

Equations (7–8) were applied to determine the mole fraction of conjugate pair from chemical shift of a specific carbon. The assignment of $^{13}C$ NMR of luteolin was further confirmed by HSQC and HMBC methods with 2D correlation with $^1H$ NMR (Supplementary Figs. 51–53).

### EPR spectroscopic measurements of flavonoids

The flavonoids solutions (5.0 mM) were loaded in a capillary tube plugged with sealing putty (Terumo Corporation, Tokyo Japan). Then the capillary tube was put into EPR tube (Wilmad Quartz (CFQ), DIAM. 5 mm), before placing in the TE mode cavity. The ESR experiments were performed at room temperature with following parameters: microwave frequency: 9.19 GHz, microwave power: 1 mW, center magnetic field: 328.348 mT, field sweep width: ±5 mT, sweep rate: 0.67 mT/min, time constant: 0.03, field modulation frequency: 500 kHz, and field modulation width: 0.025 mT. EPR data acquisition was controlled by the WIN-RAD EPR Data Analyzer System. The spectra were simulated by JEOL IsoSimu/Fa Version 2.2.0 isotropic simulation program.

**Quantitative measurement of luteolin radicals.** The area under the curve of luteolin radical EPR spectrum ($AUC_L$) are in linear relationship with the concentration of luteolin radicals, then the concentration of luteolin radical was calculated using manganese (inside sealing putty) as reference and TEMPO (25.17 mM) as standard (Eq. (9)).

$$\frac{AUC_L}{AUC_{Mn}} = \frac{AUC_{TEMPO}}{AUC_{Mn}} \quad (9)$$

**pH-dependent EPR spectrum of luteolin radicals.** Luteolin (28.6 mg, 0.1 mmol) dissolved in 10 mL KOH of different concentrations, the EPR signal of aqueous luteolin was detected immediately after transferring into EPR tube. The pH of solutions was determined after EPR measurements. Acetone and methanol were introduced to improve solubility only for samples in alkaline solution with pH < 10. Numerical data were statistically analyzed by using Origin 8.0 software package (Origin Lab, Northampton, MA). The EPR spectra of the luteolin were depicted (Supplementary Fig. 55).

### Optimized Gaussian structures of flavonoid radical anions

**Geometry optimization and spin-density distributions.** were calculated using density functional theory (DFT) at the UM062X/6-311+G(d,p) level of theory with Gaussian 09W software[37]. The cube files for both functions were generated using Multiwfn software[38] and isodensity surface plot for the non-covalent interaction studies have been

done using VMD visualization software[39]. Molecular visualizations were created using CYLview (C. Y. Legault, CYLview, 1.0b[40]).

### Monitoring of oxygen consumption rate of luteolin oxidative coupling reaction

Luteolin (214.5 mg 0.75 mmol) was added 250 mL round bottle flask. To the flask, alkaline water (50.0 mL KOH solution 0.03 M) was introduced to dissolve the luteolin and resulted in a solution with pH of 11.78. The reaction was real-time monitoring in air-tight system (Supplementary Fig. 65). The pressure changes of reaction under stirring and no stirring were recorded by pressure gauge (NVISION, pressure recorder with vacuum range of 30 MPa, CRYSTAL Engineering Corporation).

### Computational studies

**The Gibbs free energies.** were calculated using DFT computations within the Gaussian 16 program[41]. Optimizations were done based on preliminary conformational searches with Schrödinger2 Maestro 10.6. The low-energy conformers that are with 5 kcal/mol of the global minimum were re-optimized at the level of M06-2X/6-31G(d)[42], with SMD[43] solvation model for water. The vibrational frequency analyses were performed at the same level of theory to verify that minima have no imaginary frequencies and to evaluate its zero-point vibrational energy (ZPVE) and thermal corrections at 298 K. Single point energies were calculated using a larger basis set, 6-311+G(d,p), with the same solvation model.

### Antifungal activity study

The antifungal activity was evaluated against two fungi, *Aspergillus niger* ATCC 16888 and *Botrytis cinerea* ATCC 11542, isolated and identified with ITS gene (GenBank accession number AY373852)[44]. The bioassay was evaluated using the radial growth technique method[45]. Both fungal strains were cultivated aerobically on PDA medium at 25 °C for 14 days. The fungal spores were collected with sterile cotton swabs and suspended in sterile water, and the concentrations were adjusted according to the requirements of the antifungal assays to be performed. Potato Dextrose Agar (PDA) powder and dimethyl sulfoxide (DMSO) and ethanol were obtained from Merck & Co., Inc.

**Determination of in vitro antifungal activity.** Flavonoids oligomers were dissolved in a dimethyl sulfoxide (1.0 mg/mL), then transferred into sterilized warm PDA medium (40 to 45 °C) to achieve a final concentration of 1.0 mg/mL, 0.6 mg/mL and 0.4 mg/mL, before immediately pouring into 24 well sterile plastic microtitration plates containing flat-bottomed wells (Corning Incorporated, costar). After the plates were cooled to room temperature, 10 µL of freshly made *A. niger* suspension ($1.25 \times 10^{-6}$ /mL) was inoculated onto the agar of each well. Drug-free agar with 1% DMSO was used as a negative control and amphotericin B was used as positive control.

From a 7-day-old colony, the fungus with discs of 9 mm diameter was transferred to the center of the treated PDA plates and controls. All the plates were incubated at 26 ± 1 °C for 7 days. All the tested concentrations as well as positive and negative controls were measured in triplicate.

### High-throughput assay of starch hydrolase inhibition activity

The inhibitory activity of each fraction obtained in Section 2.4 on α-amylase and α-glucosidase were determined using the turbidity measurement according to a reported method[46].

### Molecular docking

**Ligand preparation.** All ligands including acarbose, luteolin, dicranolomin (**2a**), 3'''-desoxydicranolomin (**2b**) and distichumtriluteolin (**3a**)

were selected as ligands and virtually constructed through their crystal structures. These ligand molecules were drawn and saved as 3D conformers in .cif format. The structure of these ligands was then converted into .pdb format via PyMol. Subsequently, the ligand molecules were uploaded as an input file using .pdb format onto Autodock, followed by output as .pdbqt format file.

**Preparation of protein molecule.** The active center of a mammalian alpha-amylase, 1ppi, was selected and retrieved in a .pdb format file from RCSB[47]. The target amylase was loaded on the graphical user interface of Autodock[48,49] in .pdb format. The amylase was prepared for docking by removal of the acarbose molecule, deleting water molecules, adding polar hydrogen atoms and adding Kollman charges to the macromolecule. Thereafter, the amylase was converted from a .pdb format to a .pdbqt format file. A grid box was selected and adjusted to specific dimensions of the docking site. The output file of the grid dimensions was saved as a .txt file.

**Docking through AutoDock Vina and visualization using PyMOL.** AutoDock Vina[50] is a virtual screening technique to predict the optimal bound conformations of ligands to a target protein of known structure. To conduct Autodock, the prepared ligands and amylase were used in .pdbqt format and a configuration file was set up in a .txt file. Docking through AutoDock Vina was executed using command prompt and the results of the docking were analyzed through PyMOL. PyMOL is a molecular visualization program widely used for three-dimensional (3D) visualization of proteins and small molecules. The output .pdbqt file from AutoDock vina and amylase in .pdbqt format was loaded on the graphical interface of PyMOL. There, the docked structure was visualized under the molecular surface and cartoon option. The active site of the docking was shown using the pockets function of PyMOL.

**Reporting summary**

Further information on research design is available in the Nature Research Reporting Summary linked to this article.

## Data availability

1. Supplementary information include these data: (a) yields and spectroscopic data of the flavonoids synthesized; (b) ESR spectra data of the flavonoid radicals; (c) single crystal X-ray diffraction data for compounds **2a**, **2b**, **2e**, and **4a**, which were also deposited in Cambridge Data Center under deposition numbers of 2044714 (**2a**), 2044715 (**2b**), 2044716 (**2e**), and 2182228 (**4a**) and can be found at (https://www.ccdc.cam.ac.uk/structures/); (e) Semi-preparative HPLC separation conditions of the products; (f) LC-MS and [1]H NMR data for atropisomers of triflavonoids; (g) [13]C NMR data of luteolin under various pH conditions; (h) bioactivity data on alpha-amylase inhibition activity and antifungal activity; (i) molecular docking simulation data for flavone dimers with alpha-amylase; (j) Reaction condition optimization data; (k) [1]H and [13]C NMR spectra of all the synthesized compounds. 2. Supplementary data 1 includes: (a) optimized Gaussian Structures of flavonoid radical anions data; (b) Cartesian coordinated and energies of **2a**, **2a'**, and isomers of cyclotriluteolin **4a**. These data are also available from the corresponding authors upon request.

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

## Acknowledgements

The authors thank Dr Yukio Mizuta of JEOL Corporate (Japan) for help in simulating the EPR spectra of luteolin and dicranolomin radical anions. The research work was supported by the Singapore Ministry of Education grant R160-000-B04-114 and the National University of Singapore (Suzhou) Research Institute (NUSRI) via a grant under Peak of Excellent (POE) in Food Science and Technology.

## Author contributions

Conceptualization: X.Y. and D.H. Data curation: X.Y., S.H.M.L., J.L., H.T., J.Y.H.T., K.D., Y.Z., X.W., M.S., Z.S., F.L., F.Z., T.W., Ji'en Wu. Formal Analysis: X.Y. and D.H. Funding acquisition: D.H. Investigation: Experimental: X.Y., S.H.M.L., J.L., H.T., J.Y.H.T., K.D., Y.Z., X.W., M.S., and Z.S.; Computational: F.L., F.Z., and K.N.H. (DFT calculations of the Gibbs energies); T.W., S.H.M.L., C.S., X.S., and X.Y. (spin densities of radicals). Project administration: D.H. Supervision: D.H (experimental), K.N.H. (Computational). Writing—original draft: D.H. and X.Y. Writing—review & editing: K.N.H., D.H., X.Y., F.L., Z.S., S.H.M.L., J.Y.H.T., Y.K., Jie Wu.

## Competing interests

The authors declare no competing interests.
