## [Peer Review File · Nature Communications]

Oxygen Mediated Oxidative Couplings of Flavones in Alkaline WaterREVIEWER COMMENTS

Reviewer #1 (Remarks to the Author):

Due to their strong antioxidant effects, flavones represent a kind of important scavengers for radical and singlet oxygen. In present manuscript, Huang and Houk et al. described a catalyst-free oxidative coupling of luteolin and other flavones. Compared to the classic transition metal coupling reaction, this process owns good efficiency and chemical selectivity. Although the reaction requires relatively strict requirements on pKa of the substrate, a range of valuable hydroxyl-enriched bi- and tri-flavonoids were still obtained by careful selecting the substrates and conducting the reaction in specific alkaline water condition.

Mechanistic experiments revealed the radical-nucleophile coupling pathways of the reaction and uncovered the reason for optimal pH and oxygen-limiting conditions. Detailed experimental methods (such as ESR spectrum, DFT calculation) and control experiments were performed, which support the necessity of appropriate pH, catecholic groups (namely ortho-semiquinone radicals) and the unique nucleophilicity of luteolin for oxidative couplings. This reaction might inspire further exploration on synthesis of dimer and oligomer via sustainable ways. Based on above considerations, I support the manuscript to be published on nature communication after major revision. The following questions should be fully addressed.

1. The oxidative coupling reaction of luteolin with cysteine ester was firstly reported by Masuda in 2015, which was proposed a similar reaction mechanism as present manuscript except using DPPH (2,2-diphenyl-1-picrylhydrazyl). The reported result should be discussed in the introduction and the literature should be cited. (Masuda T, Nojima S, Miuraa Y, Honda S, Masuda A. An oxidative coupling product of luteolin with cysteine ester and its enhanced inhibitory activity for xanthine oxidase. *Bioorg Med. Chem. Lett.* 2015, 25(16):3117-9. doi: 10.1016/j.bmcl.2015.06.016.)

2. The authors supposed that the oxygen ($3O_2$) direct reacts as oxidant, we wondered if the real oxidant is $1O_2$, which might be generated via an energy transfer process from excited LuH_{22}^- . So, a dark experiment and the UV-vis spectrum of LuH_{22}^- is strongly suggested to be conducted.

3. Though the manuscript, it seemed that the A phenol part of flavone should have the 5,7-dihydroxyls to facilitate the reaction as nucleophile. Can resorcinol react with the produced radical? This should be further discussed and give a reasonable explanation. The limitation of current protocol should be clearly addressed.

4. Page 16 line 295-297, the author claimed that C(6) of luteolin dianion has a higher electron density. The detailed computational results were not shown in the manuscript or SI. Please give the specific data.

5. Fig 3 (c), the author proposed that isomerization could occur through ortho-semiquinone radical intermediates by hemolytic cleavage. The spin-density distribution could give some supports. Please supplement related computational study.

6. Besides using Air atmosphere (low concentration of O_2), is the reaction compatible with other similar oxidants such as peroxides, BQ, or DPPH as above-mentioned literature?

7. How about the effects of temperature and substrate concentration in this oxidative coupling process, especially for lower temperature and concentration?

8. Electrochemical experiment could be employed to exhibit the relevant characteristics of catecholic groups if possible, which may explain the formation mechanism of ortho-semiquinone radicals.

9. The heading in Figure S54, the description of corresponding relationship has some errors. The structure of C in Fig. S59 is incorrect. The authors should thoroughly correct these errors during the revision and resubmission process.

Reviewer #2 (Remarks to the Author):

Comments on the crystallography

The crystallography experimental in the supplementary information should include some details about how the solvent was modelled. The chemical formula given in the crystallographic description and accompanying table should correspond to the whole crystal, rather than just the main compound (i.e.

include the solvent).

Below are a few comments about each crystallographic sample.

Sample 2a (j459)

The differences between the reported and actual chemical formula need to be corrected.

Sample 2b (J456)

The data provided in the crystallographic description and accompanying table do not match the cif file or checkcif report, for example there are slight differences in the unit cell dimensions etc. The data is fine otherwise.

Sample 2e (J357).

There are strong indications in the checkcif alerts that the sample is twinned, namely: "PLAT906 Type_3 Report Large K values in the Analysis of Variance", "PLAT072 Type_2 Test for extreme first weighting parameter value (SHELXL)" and "PLAT931 Type_5 Test for Missed Twinning from FCF/CIF data".

The authors should check whether the rotation matrix provided in the last of these checkcif alerts is or is not a twin law. If it is, this will very likely reduce the R-factor from the present rather high value.

Sample 4a (J552)

The CCDC number provided is incorrect, so it was not possible to obtain the cif.

There is significant solvent in the voids in the lattice that is inherently difficult to model, as indicated by the large number of checkcif alerts. Without access to the cif, it is not possible to assess how many of these alerts are significant. The authors should try and resolve as many of these as is practical.

Modelling solvent is not easy, yet another look, possibly including some element of disorder, may reduce the rather high R-factor.

Reviewer #3 (Remarks to the Author):

This manuscript is an excellent one. The submission describes the dimerization and trimerization of flavones through a radical-mediated process that is confirmed using EPR. The products of the transformations are valuable natural products, and mechanistic experiments are completed to demonstrate the role of basic pH, counter ions, and oxygen. Computational analyses add insights into the interpretation of the mechanism. These reactions are, essentially, a well-developed method for the dimerization of flavones and the scope is determined. X-ray studies and 2D NMR were performed to assign the complex structures and characterize the structural rotamers. Other than a few minor corrections in the supplementary information, this submission is a breakthrough, in my opinion. However, the authors go one-step further, which presents a critical flaw. Specifically, the authors claim a 'catalyst-free' reaction has been discovered. I am hesitant to support this claim because the authors have not ruled-out the presence of a catalyst. Actually, there is not an effective experimental design that has been implemented to rigorously rule-out or characterize the presence of a catalyst. This issue consistently plagues publications on the topic of 'catalyst-free' Suzuki couplings in water that claim to not use palladium, while later studies disapprove the concept of 'catalyst-free'. As it stands, I have no choice but to evaluate the manuscript as presented; therefore, my recommendation is that the conclusions are not supported by the data in the submission.

Major Revisions:

1) The authors must add a stringent and effective research methodology to disapprove the assistance of a catalyst, even if present in small quantities. Alternatively, the author can simply remove the claim of a 'catalyst-free' transformation. The work is high impact.

Minor Revisions.

1) An important example of a pharmaceutical derived from proanthocyanidins is crofelemer (Mytesi®), which is a purified mixture of proanthocyanidins (i.e., oligomeric pro-delphinidin). This example should be added to the introduction.

2) Supplementary material: In the experimental section of 1.2.3 and 1.2.5, isolated yields are stated to be found in 1.2.8, but they are actually listed in 1.2.9. Please address this issue.

3) Supplementary material: In the experimental section of section 1.2.9, the solvent for the acquisition

of ^{13}C NMR is listed as DMSO. However, according to the spectra included in the subsequent sections, the solvent is DMSO- d_6 . Please address this discrepancy.

Reviewer #4 (Remarks to the Author):

The manuscript reports the use of a basic media in water as "catalyst" for the coupling of specific set of molecules. The molecules are rather specific, and the overall reactivity trends had been previously proposed. However, this approach had never been applied in a systematic way, and the results are nicely presented and conclusive.

DFT calculations have been used as supporting evidence for the mechanistic proposal. They rationalize the point of attack in the luteolin dianion and the isomeric preference for a specific isomer of dicranolomin. Calculations are state-of-the-art, and the agreement with experimental results lends additional support to the mechanistic proposal.

Reviewer #5 (Remarks to the Author):

The paper describes a novel catalyst-free oxidative coupling reaction of two sp^2 C-H bonds of flavones mediated by dissolved molecular oxygen as a hydrogen atom acceptor. The paper could be published subject to several major revisions as indicated below:

1. Page 4, lines 71-72: "Conducted at room temperature and food grade media (alkaline water), our reaction features high yield and good regioselectivity (Fig. 1c)."

Page 4, lines 85-87: "To verify this, we conducted HPLC analysis of the alkaline solution of luteolin (pH 11.5) and indeed found several products (Fig. S1), which were further characterized as luteolin dimers...."

It has been demonstrated in the literature the very significant effect of alkaline conditions on the stability of polyphenol compounds even at pH = 7 (J. Agric Food Chem. 2000, 48, 2101-2110; 2012, 61, 9305-9314; Food Sci. Technol. Res. (2019, 25, 123-129), This presents a serious limitation of the proposed catalyst-free coupling reaction.

2. Page 9, lines 155-161: "We dissolved apigenin, diosmetin, chrysin, wogonin, 5,6-dihydroxyflavone, and genistein in alkaline water (pH 11.5). However, no desired coupling products were detected under the same conditions. Instead, only starting materials were recovered. No free radical signals were detected by EPR spectroscopy in the reaction solution, suggesting that they are insensitive to oxygen. These flavones lack catecholic groups preventing them from forming ortho-semiquinone radical anions."

The above clearly demonstrate further disadvantages that the method is of limited applicability.

3. Page 13, lines 226-230: "Key factors influencing the reaction outcome. It is well-known that under alkaline conditions, flavonoids containing catechol moieties are sensitive to oxidation forming semiquinone radical intermediates. However, the fates of these radicals were unclear and they are not harnessed for synthetic purposes, likely due to the formation of complex end-products."

An excellent computational study (J. Phys. Chem. A, Vol. 108, No. 1, 2004, 92-96) of the bond dissociation energy and ionization potential of apigenin, luteolin, and taxifolin should be cited because a critical discussion is made on H-atom vs. electron transfer mechanism.

4. Page 14, lines 245-248: "Therefore, we determined the pKa values of specific phenolic protons by ^{13}C NMR spectra of luteolin measured under argon 29 (Fig. S48-246 S54). The first deprotonation occurred at C(7)-OH with pKa1 of 8.00. This value is about two units larger than literature values (~6.0) (20)."

The above conclusion is not correct since the cited references refer to the pKa determination under different conditions. In Ref. 28 the determination was made using methanol/water (1:2, v/v) and in Ref. 20 pKa values were determined using Britton Robin buffer / 40% ethanol (v/v). Therefore, significant differences in the pKa values are expected.

Reviewer #1 (Remarks to the Author):

Due to their strong antioxidant effects, flavones represent a kind of important scavengers for radical and singlet oxygen. In present manuscript, Huang and Houk et al. described a catalyst-free oxidative coupling of luteolin and other flavones. Compared to the classic transition metal coupling reaction, this process owns good efficiency and chemical selectivity. Although the reaction requires relatively strict requirements on pKa of the substrate, a range of valuable hydroxyl-enriched bi- and tri-flavonoids were still obtained by careful selecting the substrates and conducting the reaction in specific alkaline water condition.

Mechanistic experiments revealed the radical-nucleophile coupling pathways of the reaction and uncovered the reason for optimal pH and oxygen-limiting conditions. Detailed experimental methods (such as ESR spectrum, DFT calculation) and control experiments were performed, which support the necessity of appropriate pH, catecholic groups (namely ortho-semiquinone radicals) and the unique nucleophilicity of luteolin for oxidative couplings. This reaction might inspire further exploration on synthesis of dimer and oligomer via sustainable ways. Based on above considerations, I support the manuscript to be published on nature communication after major revision. The following questions should be fully addressed.

Reply: We highly appreciate the positive and constructive comments of the reviewers. We have taken full consideration of each comment in our revised manuscript.

1. The oxidative coupling reaction of luteolin with cysteine ester was firstly reported by Masuda in 2015, which was proposed a similar reaction mechanism as present manuscript except using DPPH (2,2-diphenyl-1-picrylhydrazyl). The reported result should be discussed in the introduction and the literature should be cited. (Masuda T, Nojima S, Miuraa Y, Honda S, Masuda A. An oxidative coupling product of luteolin with cysteine ester and its enhanced inhibitory activity for xanthine oxidase. *Bioorg Med. Chem. Lett.* 2015, 25(16):3117-9. doi: 10.1016/j.bmcl.2015.06.016.).

Reply: We noticed this nice work of luteolin coupling with cysteine. However, it is not C-C bond coupling between two flavones. Therefore, it is out of the scope of our focus. Nonetheless, we discussed such reaction briefly (page 19).

2. The authors supposed that the oxygen ($3O_2$) direct reacts as oxidant, we wondered if the real oxidant is $1O_2$, which might be generated via an energy transfer process from excited $LuH22^-$. So, a dark experiment and the UV-vis spectrum of $LuH22^-$ is strongly suggested to be conducted.

Reply: The dark experiment of luteolin-apigenin coupling reaction was conducted and the result was listed in entry No 20 f **Table S2**. The light did not involve in the reaction. The UV-VIS spectrum of $LuH2(2^-)$ did not give informative

3. Though the manuscript, it seemed that the A phenol part of flavone should have the 5,7-dihydroxyls to facilitate the reaction as nucleophile. Can resorcinol react with the produced radical? This should be further discussed and give a reasonable explanation. The limitation of current protocol should be clearly addressed.

Reply: The coupling products of luteolin and resorcinol were not observed when we react resorcinol with luteolin in reaction condition. The reasons may due to the low electron density of resorcinol in the reaction conditions.

4. Page 16 line 295-297, the author claimed that C(6) of luteolin dianion has a higher electron density. The detailed computational results were not shown in the manuscript or SI. Please give the specific data.

Reply: To elaborate the site selectivity, the averaged local ionization energy (ALIE) of Lu²⁻ is shown in Figure S82 (page 141). The lower the ALIE, the more nucleophilic it is. We revised the content in the main text accordingly (Line 294).

5. Fig 3 (c), the author proposed that isomerization could occur through ortho-semiquinone radical intermediates by hemolytic cleavage. The spin-density distribution could give some supports. Please supplement related computational study.

Reply: We amended in line 152-153 in the revised main text with an added Figure S83 showing the distribution of spin density of CTL radical.

6. Besides using Air atmosphere (low concentration of O₂), is the reaction compatible with other similar oxidants such as peroxides, BQ, or DPPH as above-mentioned literature?

Reply: We tried to conducted luteolin oxidative coupling reaction with H₂O₂ under argon protection, and we did observe luteolin dimer but only with trace amount. The H₂O₂ is not reactive as it accumulated in the reaction observed by HPLC. The reaction outcomes of luteolin with AAPH and DPPH was discussed in page 19 of the revised main text.

7. How about the effects of temperature and substrate concentration in this oxidative coupling process, especially for lower temperature and concentration?

Reply: The effects of temperatures and substrate concentration were summarised in Table S2.

8. Electrochemical experiment could be employed to exhibit the relevant characteristics of catecholic groups if possible, which may explain the formation mechanism of ortho-semiquinone radicals.

Reply: We conducted the CVs of luteolin but from the results, it is hard to give useful information regarding to semiquinone radicals, which are better characterized by the EPR spectra we have shown. Therefore, we did not include the data in the manuscript.

9. The heading in Figure S54, the description of corresponding relationship has some errors. The structure of C in Fig. S59 is incorrect. The authors should thoroughly correct these errors during the revision and resubmission process.

Reply: The errors have been corrected. Thanks for pointing out.

Reviewer #2 (Remarks to the Author):

Comments on the crystallography

The crystallography experimental in the supplementary information should include some details about how the solvent was modelled. The chemical formula given in the crystallographic description and accompanying table should correspond to the whole crystal, rather than just the main compound (i.e. include the solvent).

Below are a few comments about each crystallographic sample.

Sample 2a (j459)

The differences between the reported and actual chemical formula need to be corrected.

Reply: The formula have corrected.

Sample 2b (J456)

The data provided in the crystallographic description and accompanying table do not match the cif file or checkcif report, for example there are slight differences in the unit cell dimensions etc. The data is fine otherwise.

Reply: We have checked the table and cif file and make sure they are consistent.

Sample 2e (J357).

There are strong indications in the checkcif alerts that the sample is twinned, namely: "PLAT906 Type_3 Report Large K values in the Analysis of Variance", "PLAT072 Type_2 Test for extreme first weighting parameter value (SHELXL)" and "PLAT931 Type_5 Test for Missed Twinning from FCF/CIF data".

The authors should check whether the rotation matrix provided in the last of these checkcif alerts is or is not a twin law. If it is, this will very likely reduce the R-factor from the present rather high value.

Reply: The crystal was a two domains non-merohedral twin. Cell_Now1 was used to obtain the independent unit cells of the two domains. TwinABS2 was used for absorption correction and obtained a HKLF5 file for twin refinement. The BASF value for the twin refinement converged at 0.43461.

Sheldrick, G., Cell_Now. University of Göttingen, Germany 2008

Sheldrick, G., TWINABS, Version 2008/4. University of Göttingen, Germany 2008.

Sample 4a (J552)

The CCDC number provided is incorrect, so it was not possible to obtain the cif.

There is significant solvent in the voids in the lattice that is inherently difficult to model, as indicated by the large number of checkcif alerts. Without access to the cif, it is not possible to assess how many of these alerts are significant. The authors should try and resolve as many of these as is practical. Modelling solvent is not easy, yet another look, possibly including some element of disorder, may reduce the rather high R-factor.

Reply: We have updated the CCDC number (CCDC# 2182228).

Reviewer #3 (Remarks to the Author):

This manuscript is an excellent one. The submission describes the dimerization and trimerization of flavones through a radical-mediated process that is confirmed using EPR. The products of the transformations are valuable natural products, and mechanistic experiments are completed to demonstrate the role of basic pH, counter ions, and oxygen. Computational analyses add insights into the interpretation of the mechanism. These reactions are, essentially, a well-developed method for the dimerization of flavones and the scope is determined. X-ray studies and 2D NMR were performed to assign the complex structures and characterize the structural rotamers. Other than a few minor corrections in the supplementary information, this submission is a breakthrough, in my opinion. However, the authors go one-step further, which presents a critical flaw. Specifically, the authors claim a 'catalyst-free' reaction has been discovered. I am hesitant to support this claim because the

authors have not ruled-out the presence of a catalyst. Actually, there is not an effective experimental design that has been implemented to rigorously rule-out or characterize the presence of a catalyst. This issue consistently plagues publications on the topic of 'catalyst-free' Suzuki couplings in water that claim to not use palladium, while later studies disapprove the concept of 'catalyst-free'. As it stands, I have no choice but to evaluate the manuscript as presented; therefore, my recommendation is that the conclusions are not supported by the data in the submission.

Major Revisions:

1) The authors must add a stringent and effective research methodology to disapprove the assistance of a catalyst, even if present in small quantities. Alternatively, the author can simply remove the claim of a 'catalyst-free' transformation. The work is high impact.

Reply: We highly appreciate the encouraging remarks of our work. By "catalyst-free", we intend to highlight that the reaction do not required added catalyst (stated in line 27 of the main text) to make it work. This is in sharp contrast to many C-C bond coupling reactions that requires catalysts. Nonetheless, we replaced "Catalyst-free" with "Oxygen-Mediated" in our title.

Minor Revisions.

1) An important example of a pharmaceutical derived from proanthocyanidins is crofelemer (Mytesi®), which is a purified mixture of proanthocyanidins (i.e., oligomeric pro-delphinidin). This example should be added to the introduction.

Reply: Thank you! We have cited this important example (reference 10).

2) Supplementary material: In the experimental section of 1.2.3 and 1.2.5, isolated yields are stated to be found in 1.2.8, but they are actually listed in 1.2.9. Please address this issue.

Reply: The typo has been corrected.

3) Supplementary material: In the experimental section of section 1.2.9, the solvent for the acquisition of ¹³C NMR is listed as DMSO. However, according to the spectra included in the subsequent sections, the solvent is DMSO-d₆. Please address this discrepancy.

Reply: The typo has been corrected.

Reviewer #4 (Remarks to the Author):

The manuscript reports the use of a basic media in water as "catalyst" for the coupling of specific set of molecules. The molecules are rather specific, and the overall reactivity trends had been previously proposed. However, this approach had never been applied in a systematic way, and the results are nicely presented and conclusive.

Reply: Thank you for your positive comments!

DFT calculations have been used as supporting evidence for the mechanistic proposal. They rationalize the point of attack in the luteolin dianion and the isomeric preference for a specific isomer of dicranolomin. Calculations are state-of-the-art, and the agreement with experimental results lends additional support to the mechanistic proposal.

Reply: Thank you for your positive comments!

Reviewer #5 (Remarks to the Author):

The paper describes a novel catalyst-free oxidative coupling reaction of two sp² C-H bonds of flavones mediated by dissolved molecular oxygen as a hydrogen atom acceptor. The paper could be published subject to several major revisions as indicated below:

1. Page 4, lines 71-72: "Conducted at room temperature and food grade media (alkaline water), our reaction features high yield and good regioselectivity (Fig. 1c)."

Page 4, lines 85-87: "To verify this, we conducted HPLC analysis of the alkaline solution of luteolin (pH 11.5) and indeed found several products (Fig. S1), which were further characterized as luteolin dimers...."

It has been demonstrated in the literature the very significant effect of alkaline conditions on the stability of polyphenol compounds even at pH = 7 (J. Agric Food Chem. 2000, 48, 2101-2110; 2012, 61, 9305-9314; Food Sci. Technol. Res. (2019, 25, 123-129), This presents a serious limitation of the proposed catalyst-free coupling reaction.

Reply: The oxygen sensitivity of deprotonated polyphenolics compounds are the key to failure of coupling reactions conducted under conventional conditions with uncontrolled (often excessive) molecular oxygen in the system. We demonstrate that by controlling the oxygen availability, we can guide the reaction to form controlled oxidation products such as flavonoid dimers and trimers with good isolated yield.

2. Page 9, lines 155-161: "We dissolved apigenin, diosmetin, chrysin, wogonin, 5,6-dihydroxyflavone, and genistein in alkaline water (pH 11.5). However, no desired coupling products were detected under the same conditions. Instead, only starting materials were recovered. No free radical signals were detected by EPR spectroscopy in the reaction solution, suggesting that they are insensitive to oxygen. These flavones lack catecholic groups preventing them from forming ortho-semiquinone radical anions."

The above clearly demonstrate further disadvantages that the method is of limited applicability.

Reply: The structure and reactivity relationship of the flavones we found illustrates the scope and limitation our reaction. For those flavones that are not able to form radicals under our conditions, stronger oxidants may be used. Alternatively electrochemical mediated oxidative coupling could be applied. We have been exploring this aspect.

3. Page 13, lines 226-230: "Key factors influencing the reaction outcome. It is well-known that under alkaline conditions, flavonoids containing catechol moieties are sensitive to oxidation forming semiquinone radical intermediates. However, the fates of these radicals were unclear and they are not harnessed for synthetic purposes, likely due to the formation of complex end-products..". An excellent computational study (J. Phys. Chem. A, Vol. 108, No. 1, 2004, 92-96) of the bond dissociation energy and ionization potential of apigenin, luteolin, and taxifolin should be cited because a critical discussion is made on H-atom vs. electron transfer mechanism.

Reply: Thank you for this suggestion. We have cited the literature (reference 29). We noted that the computational study reported in this reference is conducted using neutral molecules while in our case, the oxidation reaction happens in dianion of Lu(2-) in alkaline water solution. Electron transfer, other than H abstraction is more likely to happen in our case.

4. Page 14, lines 245-248: "Therefore, we determined the pKa values of specific phenolic protons by ¹³C NMR spectra of luteolin measured under argon²⁹ (Fig. S48-246 S54). The first deprotonation occurred at C(7)-OH with pKa1 of 8.00. This value is about two units larger than literature values (~ 6.0) (20)."

The above conclusion is not correct since the cited references refer to the pKa determination under different conditions. In Ref. 28 the determination was made using methanol/water (1:2, v/v) and in Ref. 20 pKa values were determined using Britton Robin buffer / 40% ethanol (v/v). Therefore, significant differences in the pKa values are expected.

Reply: We fully agree that pKa is dependent on the solvents used. We made a comment to clarify this (line 242-243).

REVIEWER COMMENTS

Reviewer #1 (Remarks to the Author):

In present revised manuscript, all my concerns are well responded. I also evaluated carefully the author's response to other referee's comments. After revised, the manuscript has been much improved. Therefore, I am glad to recommend its publication. However, the referee suggests the authors consider using "ortho-semiquinone anion radical" instead of "ortho-semiquinone radical", which is more accurate. In addition, the phrase "in alkaline water" was suggested to add in title of the manuscript.

Reviewer #2 (Remarks to the Author):

Below are some comments on the crystallography.

Compound 4a (J552_sq, CCDC 2182228)

This structure has been refined again and the updated file sent to the CCDC. Thank you.

The

In the SOM, the phrase "A specimen of C₄₈H₃₉O₂₄ (containing three methanol and three water but 3 hydrogen cannot be located)" is ok, but we know the 3 H atoms must in the crystal, even if it is not possible to find them, so the formula with the H atoms should be used.

It should read "A specimen of C₄₈H₄₂O₂₄ (containing three methanol and three water but 3 hydrogen cannot be located)".

Related to this, in my view, it is not necessary to keep the earlier phrase "The expected formula is C₄₈H₄₂O₂₄".

The cif needs to be prepared using the correct formula, C₄₈H₄₂O₂₄, in the *.ins file and then be re-submitted to the CCDC. It is likely that checkcif will highlight that the formula does not match – in which case the authors can explain that 3 hydrogen atoms could not be located using the Validation Reply Form.

Note when I ran checkcif on the cif it says "+ solvent" included in the "moiety formula" and the "sum formula". please check this when preparing the finalised cif.

The use of PLATON and justification for its use should be included in the SOM (this info is presently only in the cif).

Please also include a discussion of how the disorder was modelled.

Compounds 2a (J459, CCDC 2044714), 2b (J456, CCDC 2044715) and 2e (J357, CCDC 2044716)

The cifs for all 3 compounds retrieved from the CCDC for review were deposited with the CCDC before the initial review of the manuscript.

In all 3 cases, there are now differences in the chemical formula reported at different parts in the Supplementary Online Material (SOM). Running checkcif on the files available from the CCDC reveal serious issues which arise due to these differences in the formula.

These differences in chemical formula suggest that there is solvent in each of the crystals that was analysed. Indeed, for one of these 3 crystal datasets solvent is mentioned in the SOM. However, the

cifs available on the CCDC for these 3 crystal structures do not have solvent in them.

Was PLATON used for these 3 structures, in a similar fashion to compound 4a? If so, there is no mention of this in the cif or the SOM?

The authors need to provide to the CCDC an updated cif for each of the 3 compounds so a review can be undertaken, and the corresponding checkcif files should be supplied for review.

Also, for J357, the authors need to indicate in the SOM that the crystal was twinned and how they handled this – their reply in the response to authors is comprehensive and can be added to the SOM.

Typographical errors

(1) for the terms R1 and wR2, the numbers should be subscript font style. Please change in both the relevant tables and the main text for all the structures reported.

(2) The authors state “where $P=(F_o2+2F_c2)/3$ ” – this should be “where $P=(F_o2+2F_c2)/3$ ”. Please change for all structures.

(3) The numbering of the crystallographic tables is presently S4, S5, S6, S7, S4, S5, S6, S7, S8, S9 etc. and needs to be corrected.

(4) For all structures, the authors state the structure solution technique was direct methods, however, the supplied cif files all state “dual space”. These are different, see Sheldrick’s article on the dual space algorithm as implemented in SHELXL:
(<https://journals.iucr.org/a/issues/2015/01/00/sc5086/index.html>)

(5) Lines 1630 and 1639, the authors state “F²”. The “2” at the end should be in the superscript font style.

(6) Lines 1632 and 1641-1642, the authors state “Å³”. The “3” at the end should be in the superscript font style.

(7) Lines 1638 and 1664, the formula should have subscript numbers (plus table S7).

(8) Line 1643, the authors state “g/cm³”. The “3” at the end should be in the superscript font style.

(9) Please check the crystallographic part of the SOM carefully for other superscript/subscript/typographical errors.

Reviewer #3 (Remarks to the Author):

The authors completed the revisions as requested.

Reviewer #5 (Remarks to the Author):

The authors have successfully incorporated the majority of the requested additions and corrections, therefore, publication is recommended.

Response to the reviewer's comments: NCOMMS-22-09812-T, 2nd revision

Reviewers' comments:

Reviewer #1 (Remarks to the Author):

In present revised manuscript, all my concerns are well responded. I also evaluated carefully the author's response to other referee's comments. After revised, the manuscript has been much improved. Therefore, I am glad to recommend its publication. However, the referee suggests the authors consider using "ortho-semiquinone anion radical" instead of "ortho-semiquinone radical", which is more accurate. In addition, the phrase "in alkaline water" was suggested to add in title of the manuscript.

Reply: Thank you for the great suggestions. We have modified the title as "Oxygen Mediated Oxidative Couplings of Flavones in Alkaline Water" and changed t ortho-semiquinone radical" with "ortho-semiquinone anion radical".

Reviewer #2 (Remarks to the Author):

Below are some comments on the crystallography.

Compound 4a (J552_sq, CCDC 2182228)

This structure has been refined again and the updated file sent to the CCDC. Thank you. Then, in the SOM, the phrase "A specimen of C₄₈H₃₉O₂₄ (containing three methanol and three water but 3 hydrogen cannot be located)" is ok, but we know the 3 H atoms must in the crystal, even if it is not possible to find them, so the formula with the H atoms should be used. It should read "A specimen of C₄₈H₄₂O₂₄ (containing three methanol and three water but 3 hydrogen cannot be located)". Related to this, in my view, it is not necessary to keep the earlier phrase "The expected formula is C₄₈H₄₂O₂₄".

The cif needs to be prepared using the correct formula, C₄₈H₄₂O₂₄, in the *.ins file and then be re-submitted to the CCDC. It is likely that checkcif will highlight that the formula does not match – in which case the authors can explain that 3 hydrogen atoms could not be located using the Validation Reply Form.

Note when I ran checkcif on the cif it says "+ solvent" included in the "moiety formula" and the "sum formula". please check this when preparing the finalised cif.

Reply: Thank you for carefully checking the cif file! It has been revised using C₄₈H₄₂O₂₄.

The use of PLATON and justification for its use should be included in the SOM (this info is presently only in the cif).

Please also include a discussion of how the disorder was modelled.

Reply: The use of PLATON and justification for its use has been included in the SOM.

Compounds 2a (J459, CCDC 2044714), 2b (J456, CCDC 2044715) and 2e (J357, CCDC 2044716)

The cifs for all 3 compounds retrieved from the CCDC for review were deposited with the CCDC

before the initial review of the manuscript.

In all 3 cases, there are now differences in the chemical formula reported at different parts in the Supplementary Online Material (SOM). Running checkcif on the files available from the CCDC reveal serious issues which arise due to these differences in the formula.

These differences in chemical formula suggest that there is solvent in each of the crystals that was analysed. Indeed, for one of these 3 crystal datasets solvent is mentioned in the SOM. However, the cifs available on the CCDC for these 3 crystal structures do not have solvent in them.

Reply: We greatly appreciate the reviewer's careful checking of the cif files. We have uploaded the revised cif files with solvent molecules and updated the CDCC numbers in SOM.

Was PLATON used for these 3 structures, in a similar fashion to compound 4a? If so, there is no mention of this in the cif or the SOM?

Reply: PLATON was not applied in these 3 structures.

The authors need to provide to the CCDC an updated cif for each of the 3 compounds so a review can be undertaken, and the corresponding checkcif files should be supplied for review.

Reply: We have upload the cif files with solvent molecules and update the CDCC numbers in SOM.

Also, for J357, the authors need to indicate in the SOM that the crystal was twinned and how they handled this – their reply in the response to authors is comprehensive and can be added to the SOM.

Reply: The description of J357 has been added to SOM. Line 1642-1643.

Typographical errors

(1) for the terms R1 and wR2, the numbers should be subscript font style. Please change in both the relevant tables and the main text for all the structures reported.

Reply: Thank you for this suggestion. The numbers have been corrected to subscript font style.

(2) The authors state "where $P=(F_o^2+2F_c^2)/3$ " – this should be "where $P=(F_o^2+2F_c^2)/3$ ". Please change for all structures.

Reply: Thank you for this suggestion. This error has been corrected.

(3) The numbering of the crystallographic tables is presently S4, S5, S6, S7, S4, S5, S6, S7, S8, S9 etc. and needs to be corrected.

Reply: Thank you for this suggestion. This error has been corrected.

(4) For all structures, the authors state the structure solution technique was direct methods, however, the supplied cif files all state "dual space". These are different, see Sheldrick's article on the dual space algorithm as implemented in SHELXL:

(<https://journals.iucr.org/a/issues/2015/01/00/sc5086/index.html>)

Reply: The Structure solution technique has been corrected from direct method to dual space.

(5) Lines 1630 and 1639, the authors state "F²". The "2" at the end should be in the superscript font style.

Reply: Thank you for this suggestion. This error has been corrected.

(6) Lines 1632 and 1641-1642, the authors state "Å³". The "3" at the end should be in the superscript font style.

Reply: Thank you for this suggestion. This error has been corrected.

(7) Lines 1638 and 1664, the formula should have subscript numbers (plus table S7).

Reply: Thank you for this suggestion. This error has been corrected.

(8) Line 1643, the authors state “g/cm³”. The “3” at the end should be in the superscript font style.

Reply: Thank you for this suggestion. This error has been corrected.

(9) Please check the crystallographic part of the SOM carefully for other superscript/subscript/typographical errors.

Reply: Thank you for this suggestion. We have checked the manuscript carefully and make sure there are no more superscript/subscript/typographical errors in it.

Reviewer #3 (Remarks to the Author):

The authors completed the revisions as requested.

Reply: Thank you for this suggestion.

Reviewer #5 (Remarks to the Author):

The authors have successfully incorporated the majority of the requested additions and corrections, therefore, publication is recommended.

Reply: Thank you for this suggestion.

REVIEWERS' COMMENTS

Reviewer #2 (Remarks to the Author):

The crystallographic comments have been addressed satisfactorily and the publication is suitable for publication.